# Vitamin K Nutrition and Bone Health

**DOI:** 10.3390/nu12071909

**Published:** 2020-06-27

**Authors:** Naoko Tsugawa, Masataka Shiraki

**Affiliations:** 1Department of Health and Nutrition, Osaka Shoin Women’s University, Higashiosaka 5778550, Japan; tsugawa.naoko@osaka-shoin.ac.jp; 2Department of Internal Medicine, Research Institute and Practice for Involutional Diseases, Nagano 3998101, Japan

**Keywords:** vitamin K requirement, vitamin K-dependent proteins, vitamin K deficiency, bone, osteocalcin, fracture

## Abstract

Vitamin K is essential for blood coagulation and plays an important role in extrahepatic metabolism, such as in bone and blood vessels, and in energy metabolism. This review discusses the assessment of vitamin K sufficiency and the role of vitamin K in bone health. To elucidate the exact role of vitamin K in other organs, accurate tools for assessing vitamin K deficiency or insufficiency are crucial. Undercarboxylated vitamin K-dependent protein levels can be measured to evaluate tissue-specific vitamin K deficiency/insufficiency. Vitamin K has genomic action through steroid and xenobiotic receptor (SXR); however, the importance of this action requires further study. Recent studies have revealed that the bone-specific, vitamin K-dependent protein osteocalcin has a close relationship with energy metabolism through insulin sensitivity. Among the organs that produce vitamin K-dependent proteins, bone has attracted the most attention, as vitamin K deficiency has been consistently associated with bone fractures. Although vitamin K treatment addresses vitamin K deficiency and is believed to promote bone health, the corresponding findings on fracture risk reduction are conflicting. We also discuss the similarity of other vitamin supplementations on fracture risk. Future clinical studies are needed to further elucidate the effect of vitamin K on fracture risk.

## 1. Introduction

Bones are structures that support locomotion through postural control and supply calcium into the blood when needed. They are equipped with mechanisms such as accumulation of calcium to maintain bone strength, and bone resorption to release calcium into the bloodstream. However, bones lose toughness with aging, leading to osteoporosis and fracture risk.

The 2000 National Institutes of Health (NIH) consensus meeting defines osteoporosis as the deterioration of bone strength [1]. Bone strength is determined by bone mineral content and bone quality and declines with abnormalities related to bone construction, such as deterioration of bone matrix proteins and bone micro-architecture, high or low bone turnover rate, and accumulation of microdamage. These abnormalities are further complicated as a result of biological senescence, which is associated with vitamin deficiencies. Vitamin D, vitamin K, and B vitamins have been known to contribute to fracture occurrence [2]. Vitamin K activates not only blood coagulation factors but also tissue-specific vitamin K-dependent proteins (VKDPs) through the posttranslational γ-carboxylation of glutamic acid (Glu) residues to γ-carboxylated glutamic acid (Gla) residues. Insufficient γ-carboxylation of VKDPs is a sensitive, tissue-specific marker of vitamin K deficiency. Moreover, vitamin K binds to the nuclear receptor steroid and xenobiotic receptor (SXR) [3]. The biological action of vitamin K through SXR includes regulation of several types of bone assembly proteins. However, there is no assay system to evaluate vitamin K activity through SXR in clinical practice.

In this review, we will examine the impact of vitamin K on bone health through influencing energy metabolism, in the light of a recent breakthrough in the understanding of the biological role of osteocalcin (OC), which is the most abundant vitamin K-dependent, bone-specific protein. The role of OC must be reviewed because OC-null mice [4] exhibited obesity and glucose intolerance, suggesting that OC may play an important role in energy and glucose metabolism as a bone-derived “hormone-like protein”.

We will also discuss the impact of vitamin K mechanisms on bone health, especially in the context of fracture risk.

## 2. Negative Correlation between Vitamin K Status and Uncarboxylated VKDPs including OC

Vitamin K plays an important role in bone metabolism by participating in the γ-carboxylation of VKDPs such as prothrombin, osteocalcin (OC), and matrix Gla protein (MGP) (Figure 1). To evaluate the nutritional and clinical significance of vitamin K, it is necessary to measure serum vitamin K homologue levels or dietary vitamin K intake. Vitamin K homologues include phylloquinone (vitamin K_1_) and menaquinones such as MK-4 and MK-7 (Figure 2). Vitamin K_1_ is abundant in green leafy vegetables because it is synthesized in the chloroplasts of plants. On the other hand, menaquinones are predominant in fermented foods because these compounds are biologically synthesized by bacteria. MK-4 is mainly present in fermented foods and meat and has lower content than vitamin K_1_ in green vegetables [5]. MK-7 is abundantly present in natto because it is synthesized by *Bacillus subtilis* var. natto. Unlike in Europeans and Americans, MK-7 is often detected in high levels in the blood of Japanese people who consume natto. Serum vitamin K homologue levels can be measured using high-performance liquid chromatography with fluorescence detection [6] or high-performance liquid chromatography–tandem mass spectrometry with atmospheric pressure chemical ionization [7]. Both methods meet high internal standards and therefore have high accuracy. Moreover, Gentili et al., Kar et al., and Riphagen et al. have developed the LC-APCI-MS/MS method that simplifies the process and shortens total run time [8,9,10,11]. Serum vitamin K homologue levels are summarized in Table 1 [8,10,12,13,14,15,16,17,18,19,20,21,22,23,24,25].

Uncarboxylated VKDPs are released into the blood circulation by various tissues that produce VKDPs. Therefore, serum levels of uncarboxylated VKDPs can be a useful marker of tissue-specific vitamin K deficiency or insufficiency. For example, protein induced by vitamin K absence II (PIVKA-II), which is an uncarboxylated form of blood clotting factor II, is specifically released by the liver and hence can be used as a marker of vitamin K deficiency in the liver. Meanwhile, uncarboxylated osteocalcin (ucOC) and uncarboxylated matrix Gla protein (ucMGP) are released by osteoblasts and blood vessels, respectively, as a result of vitamin K deficiency or insufficiency in tissues. In contrast, carboxylated osteocalcin (cOC) has an important role in bone calcification, while carboxylated matrix Gla protein (cMGP) has a strong inhibitory activity on vascular calcification.

Several studies showed a negative association between serum uncarboxylated VKDP levels and vitamin K status or dietary vitamin K intake (Table 2) [15,18,20,23,26,27,28,29,30,31,32,33,34,35,36]. However, these studies had no established criteria for detecting vitamin K deficiency in tissues. To address this, in our previous work, we established a new method for estimating vitamin K intake by curvature analysis using the serum levels of ucOC or PIVKA-II [29]. We used a logarithmic regression equation obtained from vitamin K intake values and serum ucOC or PIVKA-II levels. The cutoff point was determined to be the vitamin K intake value that showed the highest curvature. As a result, in adolescents, serum ucOC and PIVKA-II levels were negatively correlated with vitamin K intake. In the curvature analysis, the vitamin K intake values associated with bone formation and normal blood coagulation were 155–188 and 62–54 μg/day, respectively. The estimated value required for blood coagulation was approximately 1 μg/day/kg body weight, which is consistent with that of previous report that provided a basis for establishing adequate intake (AI) values of vitamin K in the dietary reference intakes (DRIs) in United States and Canada [37]. AIs of vitamin K for bone health have not been evaluated in the DRI so far, and further research is needed. The result of our study [29] inferred that the required vitamin K level for bone formation was 3 times higher than that for blood clotting, suggesting that the effect of vitamin K deficiency is more prominent on bone rather than on blood clotting in adolescents (Figure 3). A similar phenomenon was also reported in adults. Binkley et al. assessed the ability of various doses of vitamin K_1_ to facilitate osteocalcin γ-carboxylation in healthy adults aged 19–36 y. They reported that daily vitamin K intake required for γ-carboxylation of OC was >250 μg, and approximately 1000 μg/day would be required to maximally γ-carboxylate the circulating osteocalcin [18]. These vitamin K intakes required for γ-carboxylation of OC were higher than that for blood coagulation.

In addition, plasma vitamin K_1_ or MK-7 levels required to minimize the ucOC levels were highest in the ≥70 years age group, followed by the 50–69 years and 30–49 years age groups (Figure 4) [15]. The older age group may also have a higher risk of vitamin K-related vascular diseases such as aortic calcification, which has been associated with bone health [38,39]. Whether a proportion of elderly people require additional vitamin K_1_ or MK-7 for vascular disease remains unclear, and further investigation would be needed to confirm this. Further, these results [15] suggest that older people need higher levels of circulating vitamin K levels than younger people; however, it is not clear whether this phenomenon could be attributed to increased age-related requirements, or any metabolic process that is impaired owing to aging.

A negative correlation was also observed between serum ucMGP levels and vitamin K status. We confirmed that serum K_1_ levels in Japanese elderly women were negatively correlated with total ucMGP (t-ucMGP) levels and that t-ucMGP levels decreased by taking menatetrenone (MK-4) and increased by taking warfarin [33]. Regarding ucMGP, its phosphorylated and nonphosphorylated forms showed a difference in roles. It is speculated that an adequate vitamin K status would reduce the risk of vascular calcification via activation of MGP. However, further prospective intervention studies are warranted to confirm whether improvement in vitamin K status affects vascular health.

The associations between vitamin K status or uncarboxylated VKDPs and clinical outcomes are described in detail below.

## 3. Recent Topics on the Impact of Vitamin K Deficiency on Bone Health

### 3.1. Vitamin K and cOC Deficiency May Predispose to Diabetes-Related Bone Damage

In 1999, Sakamoto et al. reported that a low-vitamin K diet induced glucose intolerance in rats [40]. An intervention trial in young male volunteers also found that MK-4 increased the immunoreactive insulin response to glucose load in pre-existing vitamin K-deficient subjects [41]. Vitamin K deficiency reduced cOC and increased ucOC, whereas vitamin K repletion recovered these abnormalities [42]. Therefore, the model of cOC deficiency may mimic severe vitamin K deficiency, at least regarding OC metabolism.

Although there were no follow-up studies for Sakamoto’s studies, a breakthrough in the study of OC has been achieved by Ducy et al. [4], who were able to generate OC-null (OC-/-) mice. Their genetic modification was aimed at elucidating the role of OC in bone metabolism. The histological examination of OC-/- mouse revealed increased bone formation using the double-label technique. The absence of osteocalcin led to an increase in bone formation without impairing bone resorption [4]. However, interestingly, this genetically modified mouse exhibited other phenotypes, namely, obesity and glucose intolerance, suggesting that OC may inhibit fat accumulation and stimulate glucose metabolism [43]. Since the OC-/- mouse is considered to mimic extreme vitamin K deficiency, at least regarding OC metabolism, low serum OC levels in humans may be linked to glucose intolerance. We have previously investigated the relationship between serum OC and incident diabetes mellitus in a total of 1691 postmenopausal women belonging to the Nagano cohort study [44]. The mean observation period was 7.6 years, and 61 cases were newly diagnosed as having type II diabetes mellitus (Glyco-hemoglobin A1c (:HbA1c) was 6.5% or more, consistently) during the observation period. Table 3 indicates the age-adjusted incidence rates of diabetes mellitus with reference to the quartile with baseline OC levels. Each quartile had an insignificant difference in the observation period as expressed by person-years. As shown in the Table 3, age-adjusted hazard ratios of osteocalcin quartile to incident diabetes were 1.0 (reference population), 3.25, 3.58, and 8.05 for Q4, Q3, Q2, and Q1, respectively (*p* for trend <0.01). Thus, the lowest-OC quartile (Q1) showed 8 times higher incidence of diabetes than Q4. Moreover, in the multiple Cox proportional hazard model, presence of low osteocalcin levels (<6.1 ng/mL) was a significant and independent risk factor for type II diabetes mellitus after adjustment of confounders such as body mass index, hemoglobin A1c level, and adiponectin-to-leptin ratio (Table 4). In mouse studies, it was demonstrated that the metabolic effects of OC on glucose homeostasis are mediated through its ucOC form [43]. However, in women, only total OC, and not ucOC, levels were significantly associated with HbA1c, [45]. These inter-species differences between mouse and humans in the role osteocalcin molecule on glucose metabolism have not been resolved until now, and further studies are required in this regard.

Table 3 indicates the age-adjusted hazard ratios for type II diabetes mellitus in Q3, Q2, and Q1 in reference to Q4 (baseline OC quartile). The lowest OC quartile (Q1) showed 8 times higher incidence of diabetes than Q4 (*p* < 0.01) (Table 3).

Further adjustments for confounders for diabetes mellitus such as body mass index, hemoglobin A_1_c level, and adiponectin-to-leptin ratio, which were performed using a Cox proportional hazards model, indicated that the OC level was an independent risk factor for diabetes (Table 4).

Moreover, Pi et al. found that GPRC6A is a candidate receptor for mediating the effects of OC on insulin secretion in the pancreas [46]. These findings convey that OC is a bone-derived hormone that modulates the glucose level by influencing insulin secretion and sensitivity. Therefore, vitamin K through OC may be one of the significant regulators of glucose or energy metabolism, and we believe in the existence of a bone–pancreas relationship. In summary, vitamin K deficiency in the bone results in a lower production of OC and a low serum level of OC, predisposing to a state of glucose intolerance and diabetes mellitus that may then enhance bone matrix deterioration via the production of cross-linked advanced glycation end products (AGE) [47], which have been further associated with bone fractures after adjustment of confounders [48]. We believe that as a whole, vitamin K deficiency plays an important role in glucose metabolism, ultimately leading to a disturbance of bone quality.

### 3.2. Vitamin K Deficiency Is Associated with Bone Fractures

The relationship between vitamin K deficiency and bone fracture was first reported in patients with hip fractures [49]. Several subsequent studies also indicated that low vitamin K intake was associated with hip or vertebral fractures in both Caucasian and Asian populations [19,50,51]. Furthermore, a recent meta-analysis showed that higher vitamin K intake decreased the risk of fracture in 80,982 participants with 1114 fractures [47]. In that study, subjects with the highest vitamin K intake presented a 22% reduction in fracture risk (95% CI, 0.56–0.99). However, such anti-fracture effect of vitamin K intake was found only in groups followed up for >10 years. Other observational studies with shorter follow-up periods failed to show any significant reduction in fracture risk with higher vitamin K intake [52].

Serum vitamin K levels were also associated with bone fractures [53,54,55]. The close relationship between OC and incident fracture rate has been reported extensively. OC is synthesized by osteoblasts during the later stages of bone calcification. It is the most abundant noncollagenous protein in bone and its level in the blood is closely related to osteoblastic activity. An OC molecule generates three Glu residues in the posttranslational process, converting Glu to Gla by carboxylation (Figure 1). OC with Gla residues (cOC) tightly binds to hydroxyapatite crystals to resist against bone resorption. On the other hand, ucOC is directly secreted into the bloodstream because it has less binding ability to hydroxyapatite than cOC. In fact, compared with wild-type mice, OC knockout mice were shown to exhibit higher bone formation but faster demineralization after oophorectomy [4]. This finding may indicate that cOC is essential for keeping adequate calcium deposits after menopause. Szulc et al. first reported that serum ucOC levels were higher in patients with hip fractures than in those in the control group [56]. Tanaka et al. also found that OC levels in bone tissue were markedly lower in patients with hip fractures than in osteoarthritic patients as controls [57]. Moreover, serum ucOC levels predicted future fractures in elderly people [58,59,60]. However, there has been controversy on whether artificial vitamin K deficiency induced by warfarin use was responsible for the higher incidence of fractures. Thus, to rule out the effect of warfarin use on fracture incidence, prospective comparative studies on the relationship between vitamin K deficiency, warfarin use, and thrombin inhibitors are necessary. The comparative study on warfarin users and nonusers is hard to interpret because of many confounders.

As a whole, it is concluded that vitamin K intake and serum level of vitamin K or ucOC are associated with fractures.

## 4. Effect of Vitamin K Supplementation on Bone Turnover Rate, Bone Mineral Density, and Fractures

We previously reported that short-term vitamin K_2_ treatment decreased serum ucOC levels and increased Gla-type OC levels with modest increase in the bone resorption marker, type I collagen cross-linked N-telopeptide (NTx ) [42]. Other reports consistently indicated that vitamin K_1_ or K_2_ helped reduce serum ucOC levels, suggesting that vitamin K deficiency in bone can be addressed by vitamin K_1_ or K_2_ administration without any change in bone mineral density (BMD) at any site of bone tissue [61,62,63,64]. A meta-analysis revealed that modest overall treatment effects of vitamin K on BMD with significant bias were observed [65]. Therefore, the effects of vitamin K on BMD should be interpreted with caution. Several other studies also showed unclear findings on the effect of vitamin K treatment on fracture risk [66,67]. In a recent meta-analysis, vitamin K_2_ played a specific role in the maintenance of vertebral BMD and the prevention of fractures [67]. Another meta-analysis reported that vitamin K may reduce the rate of bone loss, but the reduction in fracture risk was observed only in a Japanese population [66]. Therefore, these studies commonly suggest that the effect of vitamin K on BMD may be minimal but that vitamin K deficiency in bone could be improved by vitamin K treatment because the results consistently showed reduced serum ucOC levels. The effect of vitamin K on fracture risk has not been established yet.

Interestingly, these effects of vitamin K on bone health are similar to those of vitamin D or the vitamin B group. Vitamin D deficiency has been reported to increase fracture susceptibility [68,69]. Meanwhile, vitamin B deficiency is a risk factor for fracture susceptibility, independent of BMD [70,71], because such deficiency induces high levels of homocysteine. However, the supplementation of either vitamin resulted in no beneficial effect on fracture prevention [72,73]. It is well known that serum 25(OH) vitamin D levels are increased by the administration of native vitamin D and that serum homocysteine levels are decreased by the administration of vitamin B6, B12, and folate; similarly, serum levels of ucOC were also reduced by vitamin K administration, suggesting that vitamin B, D, and K deficiencies are treatable with the corresponding vitamin administration. If a deficient vitamin is improved by the administration of the corresponding vitamin, the phenotype of vitamin deficiency is also expected to recover.

As mentioned above, the literature shows no consistent evidence on the beneficial effects of vitamin supplementation on fracture prevention. The plausible reasons of the contradictory results could be the following. Long-term observational studies showed that fracture susceptibility in vitamin-deficient subjects increased after around 10 years of observation, but studies investigating the effect of vitamin exposure on fracture were carried out for only 2–3 years. Another reason could be the very small statistical power for determining the effect of each vitamin deficiency on fracture susceptibility [2]. In fact, since people usually consume foods low in many vitamins, correcting a single vitamin deficiency is not enough to prevent fractures [2]. Therefore, a more accurate adjustment for confounders may be required when exploring the effects of vitamin K on fracture prevention. Furthermore, as vitamin K has additional biological effects on bone, such as improvement of bone geometry [74] and genomic action [3], their impact on fracture reduction also needs further clarification.

## 5. Possible Implications of Combined Vitamin D and K Supplementation on Bone Quality

Since the gene expression of OC is enhanced by 1,25(OH)_2_D following which the OC molecule receives a posttranslational modification by vitamin K [75], it can be hypothesized that a synergistic or additive effect of vitamins D and K may exist in the bone. Further, an in vitro study indicated that vitamin K_2_ promoted 1 alpha,25(OH)_2_D_3_-induced mineralization in human periosteal osteoblasts [76]. Furthermore, a study on human osteoblast cell culture demonstrated that glycoxidation interfered with maturation of osteoblasts, and this process was counterbalanced by adding vitamins D and K that reversed the detrimental effect of pentosidine, which is a well-characterized cross-linking, advanced AGE [77]. We have previously reported that multiple-vitamin deficiency carried a higher fracture risk than single-vitamin deficiency [2]. Based on these reports, it can be inferred that supplementation of vitamins D and K together might have a greater beneficial effect on bone quality than that obtained by their individual supplementation, especially during the recovery of osteoblasts from the effects of accumulated AGEs.

## 6. The Role of MGP (Matrix Gla Protein) in Bone Health

Matrix Gla protein (MGP) is a secreted extracellular protein that is initially extracted and purified from demineralized bone. MGP had been reported to inhibit tissue mineralization because the MGP-knock out mice revealed marked vascular and cartilage calcification [78]. In human study, Lawton et al. reported that the osteoblasts from normally healing fractures did not exhibit MGP. However, the osteoblasts isolated from a non-union site exhibited positive for MGP, suggesting that MGP may interfere with normal calcification of bone during fracture healing [79]. However, the following in vitro studies expanded the physiological role of MGP on bone metabolism. Zhang et al. reported that MGP promotes bone formation [80] by upregulating Wnt/β-catenin signaling pathway and inhibits osteoclasts differentiation [81]. These two findings indicated the possible role of MGP on the pathogenesis of osteoporosis.

The complexly of MGP can be explained by the complex posttranslational modification of MGP. This molecule required not only carboxylation at five Glu residues (ucMGP) to Gla (cMGP) but also phosphorylation at three serine residues (pMGP). Phosphorylation was essential for the secretion of MGP into blood stream and requires phosphate. Vitamin K deficiency leads to a decrease in the levels of the phosphorylated carboxylated MGP (p-cMGP) and a rise in the levels of dephosphorylated undercarboxylated MGP (dp-ucMGP) [82], especially in patients with stage 5 chronic kidney disease. Further clarification of the role of vitamin K deficiency on the pathogenesis of various diseases such as osteoporosis, osteoarthrosis, or vascular calcification will be required in view of MGP metabolism.

## 7. Conclusions

Assessing vitamin K deficiency or insufficiency can help explain the role of vitamin K in extrahepatic organs including bone tissue. To assess vitamin K deficiency or insufficiency, it is necessary to use accurate tools such as high-performance liquid chromatography–tandem mass spectrometry with atmospheric pressure chemical ionization to measure serum vitamin K levels. Measuring undercarboxylated vitamin K-dependent protein (i.e., osteocalcin) levels is also considered the most accurate and convenient method for assessing tissue-specific vitamin K deficiency or insufficiency. This bone-specific vitamin K protein has been reported to influence glucose metabolism; moreover, it is structurally similar to bone-derived hormone. This review also revealed that the association between vitamin K deficiency and fracture risk has been established by many epidemiological studies. However, until now, findings on the effects of vitamin K supplementation on fracture risk remain inconclusive; thus, further studies regarding this aspect are required. The same observations have been noted for vitamins D and B, because their deficiencies are reported to be risks for fracture, but their supplementation did not show clear risk reduction. The lack of adequate clinical studies, especially in regard to fracture prevention, is a limitation of the present review. Therefore, further clarifications in the context of confounder adjustment and longer follow-up periods are required to help better understand the effect of vitamin supplementation on fracture risk. A recent in vitro study revealed that a combined vitamin D and K supplementation might help in recovery of bone quality that is otherwise not treatable by the traditional drugs for osteoporosis. This possibility should also be examined more extensively. Matrix Gla protein, which is another important vitamin K-dependent protein in bone, cartilage, and vessels, is activated by posttranslational modification on Glu and Ser residues of MGP by carboxylation and phosphorylation, respectively. The recent studies indicated that MGP may interact with calcification of osteoblasts and osteoclasts differentiation. It is important to elucidate the role of MGP on bone and vessel health in future.

## Figures and Tables

**Figure 1 nutrients-12-01909-f001:**
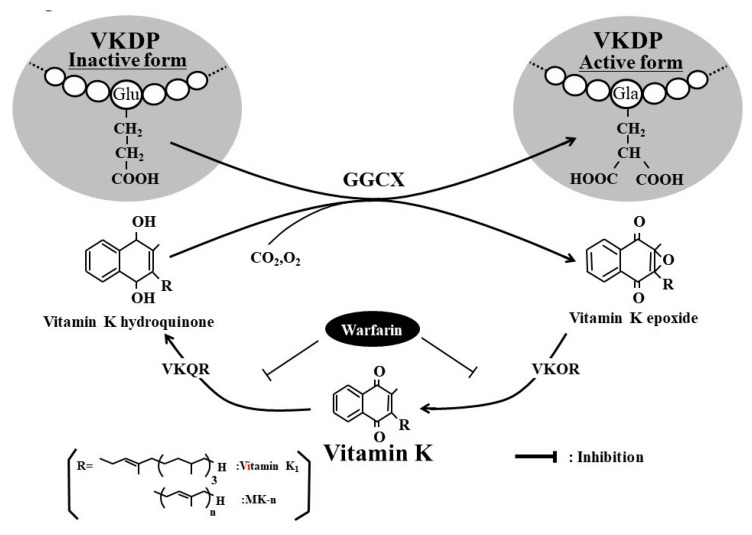
Activation of vitamin K-dependent protein (VKDP) by vitamin K and γ-glutamyl carboxylase (GGCX). Vitamin K is a cofactor for GGCX, which is responsible for converting specific glutamic acid (Glu) residues to γ-carboxylated glutamic acid (Gla) residues in VKDPs such as prothrombin, osteocalcin (OC), and matrix Gla protein (MGP). Vitamin K plays a role in normal blood coagulation via the γ-carboxylation of coagulation factors (II, VII, IX, and X), increases bone strengthen via the γ-carboxylation of OC, and suppresses arterial calcification via the γ-carboxylation of MGP. In vitamin K insufficiency or deficiency, undercarboxylated VKDP, an inactive form of VKDP, is released from the target organ. Thus, serum levels of undercarboxylated VKDPs are a sensitive marker of vitamin K status. VKOR, vitamin K epoxide reductase; VKQR, vitamin K quinone reductase.

**Figure 2 nutrients-12-01909-f002:**
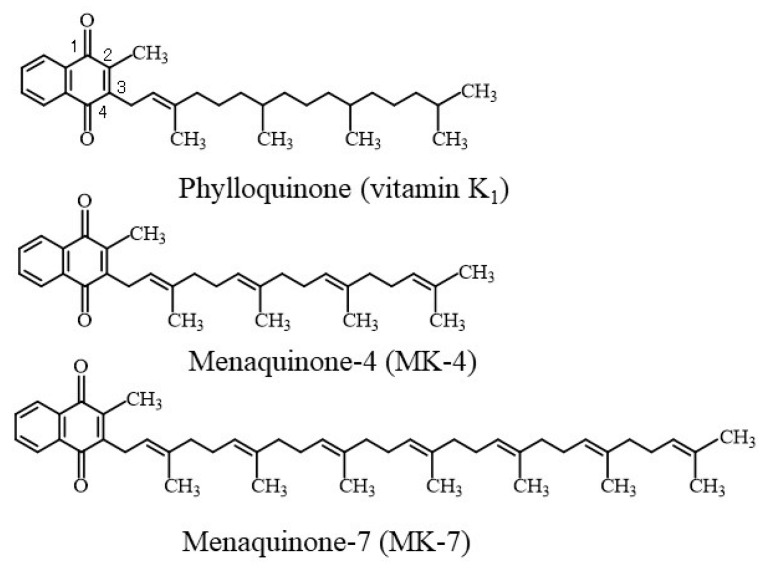
Chemical structures of vitamin K homologues. Vitamin K naturally exists in two forms, namely, phylloquinone (vitamin K_1_) and menaquinones (MK-n, or vitamin K_2_). Vitamin K homologues are characterized by a 2-methyl-1,4-naphthoquinone nucleus and a polyisoprenoid side chain at the 3-position that varies in both length and degree of saturation.

**Figure 3 nutrients-12-01909-f003:**
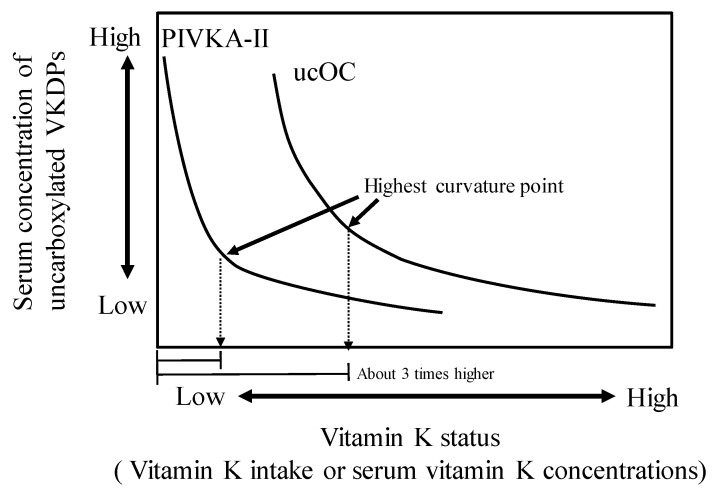
Relationship between vitamin K nutritional status and uncarboxylated vitamin K-dependent proteins (VKDPs) as a vitamin K deficiency marker in adolescents (adapted from reference [29]). Abbreviations: PIVKA-II, protein induced by vitamin K absence or antagonist-II; ucOC, uncarboxylated osteocalcin.

**Figure 4 nutrients-12-01909-f004:**
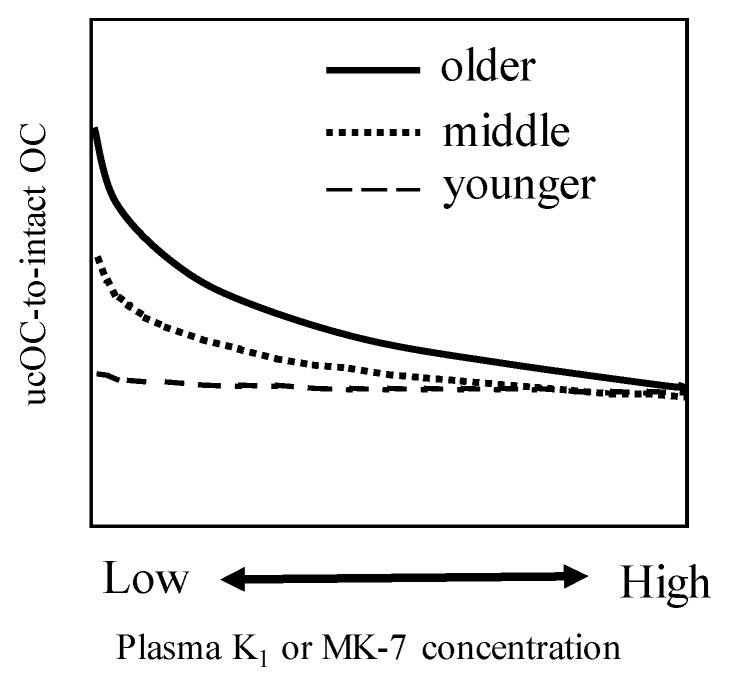
Relationship between plasma vitamin K_1_ or menaquinone-7 (MK-7) levels and uncarboxylated osteocalcin (ucOC)-to-intact osteocalcin (OC) ratio according to age group. Older; >=70 years old, middle; 50–69 years old, younger; 30–49 years old (adapted from reference [15]).

**Table 1 nutrients-12-01909-t001:** Phylloquinone (vitamin K_1_) and menaquinone-7 (MK-7) levels in the blood circulation.

Authors (Year of Publication)	Subjects	Mean Age (Years)	Average Serum or Plasma Level (nmol/L)		Country	Reference
			Vitamin K_1_	MK-7		
Nakano et al.	Elderly men	82.2	1.22	6.6	Japan	[12]
(2011)	Elderly men with fractures	82.6	0.69	2.5		
	Elderly women	84.1	1.71	16.6		
	Elderly women with fractures	85.5	1.02	4.1		
Kuwabara et al.	Institutionalized elderly men	84.9	1.38	1.2	Japan	[13]
(2010)	Institutionalized elderly women	88.7	1.31	0.8		
Martini et al. (2006)	Postmenopausal women	70-71	1.7–1.8	*-*	USA	[14]
Tsugawa et al.	30–49 years-old women	45.4	3.37	7.6	Japan	[15]
(2006)	50–69 years-old women	59.6	3.86	13.0		
	70 years-old women	74.9	2.86	6.5		
Beavan et al.	Postmenopausal women (Chinese)	67.6	2.22	*-*	UK	[16]
(2005)	Postmenopausal women (British)	67.4	0.69	*-*		
	Postmenopausal women (Gambian)	66.7	0.80	*-*		
Booth et al. (2004)	Adult men (vitamin K intake, 151 μg/day)	59	1.54	*-*	USA	[17]
	Premenopausal women (vitamin K intake, 17 μg/day)	47.3	1.05	*-*		
	Postmenopausal women (vitamin K intake, 177 μg/day)	63	1.41	*-*		
Binkley et al. (2002)	Young adult men and women (vitamin K intake, 77–120 μg/day)	25	0.61–1.0	*-*	USA	[18]
Binkley et al. (2002)	Subjects with vitamin K_1_ intake of 375–500 μg/day		2–4	*-*	USA	[18]
Kaneki et al.	Postmenopausal women (Tokyo)	57.2	1.61	8.10	Japan	[19]
(2001)	Postmenopausal women (Hiroshima)	67.4	1.64	1.88		
Binkley et al.	Young adult men	24	0.68	*-*	USA	[20]
(2000)	Young adult women		0.72	*-*		
	Elderly men	75	1.03	*-*		
	Elderly women		1.16	*-*		
Hodges et al.	Elderly women with fractures	81.7	0.75	0.18	France	[21]
(1993)	Healthy elderly women	80	1.30	0.35		
Gentili A et al.	Healthy subjects (*n* = 5)		0.947		Italy	[8]
(2014)	Patients under oral anticoagulant therapy (*n* = 5)		0.691			
Riphagen et al. (2015)	Renal transplant recipients (*n* = 60)	55	1.35 (0.89–2.32)	<4.40	Netherlands	[10]
Fusaro et al. (2012)	Healthy subjects (*n* = 62)	56.8	1.36	2.53	Italy	[22]
	Dialysis paients (*n* = 387)	64.2	0.98	0.87		
Holden et al. (2010)	Chronic kidney disease stages 3–5 (average vitamin K intake, 130 µg/day; 17.3–740 µg/day)	61	2.1 (0–19.3)	-	Canada	[23]
Pilkey et al. (2007)	Dialysis patients	62.6	0.99	-	Canada	[24]
Holden et al. (2008)	Peritoneal dialysis patients (average dialysis period, 49 months)	56 (28.7–85)	Median, 0.7 (0.1–2.2)	-	Canada	[25]

K_1_, phylloquinone; MK-7, menaquinone-7; -, not measured; USA, United States of America; UK, United Kingdom. All vitamin K levels were expressed in unit of nmol/L according to the molecular weight (K_1_: 450.7 g mol^-1^, MK-7: 649.0 g mol^-1^).

**Table 2 nutrients-12-01909-t002:** Relationship between uncarboxylated vitamin K dependent proteins (VKDPs) and vitamin K status.

Uncarboxylated VKDP	Target Tissue	Subjects	Relationship to VK Status	Reference (Year)	
PIVKA-II	Liver	Adult men and women	Negative correlation with serum VK_1_ levels	Sokoll et al. (1996)	[26]
		Elderly women	Increased by VK intake restriction; decreased by VK_1_ supplementation (86 μg/day)	Booth et al. (2003)	[27]
		Adult patients with chronic kidney disease (stages 3–5)	Negative correlation with VK intake	Holden et al. (2010)	[23]
		Elderly men and women	Negative correlation with VK intake	Kuwabara et al. (2011)	[28]
		Adolescent boys and girls	Negative correlation with VK intake; required intake levels were at least 62 μg/day for boys and at least 54 μg/day for girls (approximately 1 μg/day/kg body weight)	Tsugawa et al. (2012)	[29]
		Dialysis patients aged >18 years	Decreased by MK-7 supplementation	Westenfeld et al. (2012).	[30]
ucOC	Bone	Young and elderly men and women	Negative correlation with serum VK_1_ levels	Sokoll et al. (1996)	[26]
		Young and elderly men and women	Negative correlation with serum K_1_ levels; decreased by VK_1_ supplementation	Binkley et al. (2000)	[20]
		Healthy adults	Decreased by VK_1_ supplementation	Binkley et al. (2002)	[18]
		Elderly women	Increased by VK intake restriction; decreased by VK_1_ supplementation	Booth et al. (2003)	[27]
		Elderly women	Negative correlation with serum VK levels	Tsugawa et al. (2006)	[15]
		Young adult men and women	Positive correlation between cOC-to-ucOC ratio and VK_1_ (MK-7) supplementation	Schurgers et al. (2007)	[31]
		Elderly men and women	Negative correlation between VK intake and OCR	Kuwabara et al. (2011)	[28]
		Dialysis patients aged >18 years	Decreased by MK-7 supplementation	Westenfeld et al. (2012).	[30]
		Adolescent boys and girls	Negative correlation with VK intake; required VK intake levels were 155–188 μg/day	Tsugawa et al. (2012)	[29]
t-ucMGP	Blood vessels	Men and women in their 50s with hypertension	Positive correlation with OCR	Rennenberg et al. (2010)	[32]
		Elderly women	Negative correlation with serum K_1_ level; decreased by taking menatetrenone (MK-4); increased by taking warfarin	Tsugawa et al. (2014)	[33]
dp-ucMGP	Blood vessels	Adults	Decreased by VK intake; increased by taking warfarin	Schurgers et al. (2008)	[34]
		Elderly men and women	Negative correlation with VK intake and serum PK levels; positive correlation with PIVKA-II levels and %ucOC	Shea et al. (2011)	[35]
		Dialysis patients aged >18 years	Decreased by MK-7 supplementation	Westenfeld et al. (2012).	[30]
		Elderly women	Positive correlation with OCR (no correlation between OCR and dp-cMGP or t-ucMGP)	Dalmeijer et al. (2013)	[36]

PIVKA-II, protein induced by vitamin K absence or antagonist-II; ucOC, uncarboxylated osteocalcin; t-ucMGP, total uncarboxylated matrix Gla protein; dp-ucMGP, desphospho-uncarboxylated matrix Gla protein; VK, vitamin K; VK_1_, phylloquinone; MK-7, menaquinone-7; OCR, uncarboxylated osteocalcin-to-carboxylated osteocalcin ratio; cOC, carboxylated osteocalcin; %ucOC, percentage of uncarboxylated osteocalcin to total osteocalcin.

**Table 3 nutrients-12-01909-t003:** Age-adjusted incidence rates of diabetes mellitus with reference to the quartile with baseline osteocalcin levels. (Adapted from [44]).

Quartile by Osteocalcin Level	Observation Period (Person-Years)	Number of Incident DM Cases	Incident Rate (per 1000 Person-Years)	Age-Adjusted HR (95% CI)	*p* vs. Q4
Q4	3293	4	1.2	1.00 (reference)	
Q3	3183	13	4.1	2.25 (1.15–11.6)	<0.05
Q2	3324	14	4.2	3.58 (1.28–12.6)	<0.01
Q1	3121	30	9.6	8.05 (3.17–27.1)	<0.01

DM, diabetes mellitus; HR, hazards ratio; CI, confidence interval. Q1 represents the quartile with the lowest serum osteocalcin levels. Q2, Q3 and Q4 represent the secondary, tertiary and the highest quartiles, respectively.

**Table 4 nutrients-12-01909-t004:** Multivariate Cox proportional hazard model for incidence of type 2 diabetes mellitus adjusted for confounders. (Adapted from [44]).

Item	HR	95% CI	*P*
Age, years	1.049	1.003–1.096	0.037
BMI, kg/m^2^	1.078	0.974–1.193	0.149
Osteocalcin, <6.1 ng/mL	2.481	1.274–4.833	0.008
Triglycerides, mg/dL	1.001	0.998–1.004	0.376
NTX, nmol /nmolCr	0.999	0.983–1.015	0.911
hs-CRP, mg/dL,	1.227	0.285–5.287	0.784
Adiponectin/leptin ratio	0.803	0.644–1.000	0.050
HbA_1_c, %	2.518	1.858–3.414	<0.0001
L2-4BMD, g/cm^2^	1.497	0.306–7.330	0.619
Phosphate, mg/dL	1.908	0.910–4.003	0.087
Homocysteine, nmol/mL	1.017	0.918–1.127	0.745

HR: hazard ratio, CI: confidence interval, BMI: body mass index, NTX: N-teropeptide, hs-CRP: high-sensitivity C-reactive protein, L2-4BMD: lumbar (L2–4) bone mineral density. HbA1c: glycohemoglobin A1c.

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
