# Peer review of "Vitamin K Nutrition and Bone Health"

_nutrients, 2020, doi:10.3390/nu12071909_

Round 1
Reviewer 1 Report
As I commented before, there is one similar review which just was published in Nutrients. This work does not have a scientific contribution to the field.
Author Response
< Comment of reviewer 1 and our response>
(Reviewer’s comment)
As I commented before, there is one similar review which just was published in Nutrients. This work does not have a scientific contribution to the field.
(Response)
We disagreed to the reviewer’s comment.
We have checked the review articles published in Nutrients during recent several years. We found a total of four articles, in which stated on vitamin K nutrition.
1. MK-7 and its effects on bone quality and strength. Nutrients 2020: 12(4): 965.
2. Vitamin K as a diet supplementation with impact human health. Current evidence in age-related diseases. Nutrients 2020: 12(1).
3. Undercarboxylated osteocalcin: Experimental and human evidence for a role in glucose homeostasis and muscle regulation of insulin sensitivity. Nutrients 2018: 10 (7).
4. Vitamin K2 therapy for postmenopausal osteoporosis. Nutrients 2014: 6(5).
First of all, we had not participated the presentation mentioned above. In addition, these 4 articles included the different contents comparing to our present review. In fact, the reference No 1 claimed that MK-7 had a most potent biological activity to covert Glu residue to Gla in osteocalcin molecule. On the other hand, we summarized the effect of MK-4 on fracture prevention. Thus, the contents of the article and those in ours were quite different.
The reference No. 2 was mainly stated vitamin K bioavailability and the determination procedure of vitamin K RDA was different from ours. Ref 3 was stated about the role of osteocalcin molecule on energy metabolism. This topic was also mentioned in our present review. However, we have pointed out that the role of osteocalcin molecule on glucose metabolism was different between rodent model and human studies. Because the essential molecule of osteocalcin in rodent was ucOC but in humans, the role of ucOC on glucose metabolism was conflict. We expect the difference in osteocalcin molecule between rodent and humans may result in the vitamin K sufficiency.
The ref No 3 was simply introduced the article written by Lee et al (Cell) without any objection between rodent and humans. Lee et al. indicated that ucOC in rodent was produced by decarboxylation through bone resorption. However, the ucOC in humans is produced by vitamin K deficiency. Therefore, the ucOC in humans did not relate with glucose metabolism. We have shown the role of intact osteocalcin molecule on the occurrence of type II diabetes by the retrospective observational study in postmenopausal women. This evidence was the first evidence to show the tight relationship between incident diabetes and the baseline osteocalcin molecule in humans using retrospective long-term observational study. Therefore, the ref No 3 and ours were again different.
Ref No 4 was summarized the clinical trials to explore the effect of vitamin K on fracture prevention using the articles published until 2014. We have followed another 3 meta-analysis publications. Those were concluded the effect of vitamin K on fracture prevention were still controversial. We added the possible reasons why vitamin K2 did not indicate the fracture prevention, persistently. Thus, we believe that the conclusions and discussions between these antecedent articles and our present review were different and ours includes novel views.

Reviewer 2 Report
The manuscript had required extension revision and I am satisfied with how the authors clarified each point.
Author Response
Thank you for acknowledging our revised manuscript.
Reviewer 3 Report
The paper by Tsugawa and Shiraki:Vitamin K Nutrition and Bone Health, It’s interesting enough but it needs revision:
- page 2 line between 67-69: The method for assessing vitamin K is not standardized and The Authors should specify this issue adding the following manuscripts: Gentili A et al, Rapid, high performance method for the determination of vitamin K(1), menaquinone-4 and vitamin K(1) 2,3-epoxide in human serum and plasma using liquid chromatography-hybrid quadrupole linear ion trap mass spectrometry. J Chromatogr A. 2014 Apr 18;1338:102-10 - Fusaro et al. Vitamin K plasma levels determination in human health. Clin Chem Lab Med. 2017 May 1;55(6):789-799.
- Page 4 End of the Table 1 (from Holden et al): the Authors show in table 1, three studies with very poor statistical power and they didn't cite two relevant studies because they evaluate clinical outcomes (Fractures, Vascular calcifications and mortality) such as: Evenepoel P et al, Poor Vitamin K Status Is Associated With Low Bone Mineral Density and Increased Fracture Risk in End-Stage Renal Disease,J Bone Miner Res. 2019 Feb;34(2):262-269 and Fusaro M et al, Vitamin K, Vertebral Fractures, Vascular Calcifications, and Mortality: VItamin K Italian (VIKI) Dialysis Study, J Bone Miner Res. 2012 Nov;27(11):2271-8. The Authors are strongly advised to add both the studies (Evenepoel et al and Fusaro et al) in Table 1.
- The paragraph page 8: Recent Topics on the impact of Vitamin K Deficiency on Bone Heath the Authors describe correlation between BGP levels and Diabetes. First of all this Topic is not recent and furthermore human data are very uncertain. The Authors are strongly advised to describe briefly novelty aspect related to BGP and its role to prevent Vascular Calcifications and recent role of MGP to prevent Bone Fractures adding the following citations: Zhang Y et al, Unexpected Role of Matrix Gla Protein in Osteoclasts: Inhibiting Osteoclast Differentiation and Bone Resorption. Mol Cell Biol. 2019 May 28;39(12):e00012-19. and Fusaro et al, Correction to: Vitamin K effects in human health: new insights beyond bone and cardiovascular health.J Nephrol. 2020 Apr;33(2):389
Author Response
<Comments of reviewer 3 and our responses>
Comment 1.
Page 2 line between 67-69: The method for assessing vitamin K is not standardized and The Authors should specify this issue adding the following manuscripts: Gentili A et al, Rapid, high performance method for the determination of vitamin K(1), menaquinone-4 and vitamin K(1) 2,3-epoxide in human serum and plasma using liquid chromatography-hybrid quadrupole linear ion trap mass spectrometry. J Chromatogr A. 2014 Apr 18;1338:102-10 - Fusaro et al. Vitamin K plasma levels determination in human health. Clin Chem Lab Med. 2017 May 1;55(6):789-799.
(Response)
Thank you for your helpful suggestion. In accordance with reviewer’s suggestion, we added a sentence as follows in line 68-70. And also, we added the studies related to LC-APCI-MS/MS for vitamin K detection in addition to those suggested by the reviewer.
Line 68- “Moreover, Gentili et al, Kar et al, Riphagen et al have developed the LC-APCI-MS/MS method that simplified in the process and shorten total run-time [8-11].”
[References]
- Gentili, A.; Cafolla, A.; Gasperi, T.; Bellante, S.; Caretti, F.; Curini, R.; Fernández, V.P. Rapid, high performance method for the determination of vitamin K(1), menaquinone-4 and vitamin K(1) 2,3-epoxide in human serum and plasma using liquid chromatography-hybrid quadrupole linear ion trap mass spectrometry. J. Chromatogr. A. 2014. 1338, 102-110.
- Karl, J.P.; Fu, X.; Dolnikowski, G.G.; Saltzman, E.; Booth, S.L. Quantification of phylloquinone and menaquinones in feces, serum, and food by high-performance liquid chromatography-mass spectrometry. J. Chromatogr. B Analyt. Technol. Biomed. Life. Sci. 2014, 963, 128-133.
- Riphagen, I.J.; van der Molen, J.C.; van Faassen, M.; Navis, G.; de Borst, M.H.; Muskiet, F.A.; de Jong, W.H.; Bakker, S.J.; Kema, I.P. Measurement of Plasma Vitamin K1 (Phylloquinone) and K2 (menaquinones-4 and -7) Using HPLC-tandem Mass Spectrometry. Clin. Chem. Lab. Med. 2016, 54, 201-210.
- Fusaro, M.; Gallieni, M.; Rizzo, M.A.; Stucchi, A.; Delanaye, P.; Cavalier, E.; Moysés, R.M.A.; Jorgetti, V.; Iervasi, G.; Giannini, S.; Fabris, F.; Aghi, A.; Sella, S.; Galli, F.; Viola, V.; Plebani, M. Vitamin K plasma levels determination in human health. Clin. Chem. Lab. Med. 2017, 55, 789-799.
Comment 2.
Page 4 End of the Table 1 (from Holden et al): the Authors show in table 1, three studies with very poor statistical power and they didn't cite two relevant studies because they evaluate clinical outcomes (Fractures, Vascular calcifications and mortality) such as: Evenepoel P et al, Poor Vitamin K Status Is Associated With Low Bone Mineral Density and Increased Fracture Risk in End-Stage Renal Disease,J Bone Miner Res. 2019 Feb;34(2):262-269 and Fusaro M et al, Vitamin K, Vertebral Fractures, Vascular Calcifications, and Mortality: VItamin K Italian (VIKI) Dialysis Study, J Bone Miner Res. 2012 Nov;27(11):2271-8. The Authors are strongly advised to add both the studies (Evenepoel et al and Fusaro et al) in Table 1.
(Response)
Thank you for your useful suggestion. In accordance with reviewer’s suggestion, vitamin K1 concentrations of healthy subjects and dialysis patients in the study of Fusaro et al [ref.No.22] were added in Table 1. However, we could not refer the study of Evenepoel et al, because they did not mention the vitamin K concentration in their study that reviewer have suggested.
Comment 3.
The paragraph page 8: Recent Topics on the impact of Vitamin K Deficiency on Bone Heath the Authors describe correlation between BGP levels and Diabetes. First of all this Topic is not recent and furthermore human data are very uncertain. The Authors are strongly advised to describe briefly novelty aspect related to BGP and its role to prevent Vascular Calcifications and recent role of MGP to prevent Bone Fractures adding the following citations: Zhang Y et al, Unexpected Role of Matrix Gla Protein in Osteoclasts: Inhibiting Osteoclast Differentiation and Bone Resorption. Mol Cell Biol. 2019 May 28;39(12):e00012-19. and Fusaro et al, Correction to: Vitamin K effects in human health: new insights beyond bone and cardiovascular health. J Nephrol. 2020 Apr;33(2):389
(Response)
Thank you for your constructive advice. In accordance with your advice we added the new section, in which focused on recent discoveries of role of MGP on bone.
We added the following sentences to deal with this topic in section 6.
- The role of MGP (Matrix gla protein) in bone health.
Matrix Gla protein (MGP) is a secreted extracellular protein that is initially extracted and purified from demineralized bone. The role of MGP had been reported to inhibit tissue mineralization because the MGP null-mouse revealed marked vascular and cartilage calcification [77]. In human study, Lawton et al. reported that the osteoblasts from normally healing fractures did not exhibit MGP. However, the osteoblasts isolated from non-union site exhibited positive for MGP, suggesting that MGP may interfere normal calcification of bone during fracture healing [78]. However, the following in vitro studies expanded the physiological role of MGP on bone metabolism. Zhang et al reported that MGP promotes bone formation [79] by up-regulating Wnt/-catenin signaling pathway and inhibit osteoclasts differentiation [80]. These two findings indicated the possible role of MGP on the pathogenesis of osteoporosis.
The complexly of MGP can be explained by the complex posttranslational modification of MGP. This molecule required 5 carboxylation at Glu residues (ucMGP) to Gla (cMGP) and phosphorylation at 3 serine residues (pMGP). Phosphorylation was essential for the secretion of MGP into blood stream and requires phosphate. Vitamin K deficiency leads to a decrease in the levels of the phosphorylated-carboxylated MGP (p-cMGP) and a rise in the levels of dephosphorylated undercarboxylated MGP (dp-ucMGP) [81], especially in patients with stage 5 chronic kidney disease. Further clarification of the role of vitamin K deficiency on the pathogenesis of various diseases such as osteoporosis, osteoarthrosis or vascular calcification will be required in view of MGP metabolism.
The following sentences were added into the conclusion section.
Matrix Gla protein, which is other important vitamin K dependent protein in bone, cartilage and vessels, is activated by posttranslational modification on Glu and Ser residues of MGP by carboxylation and phosphorylation, respectively. The recent studies indicated that MGP may interact with calcification of osteoblasts and osteoclasts differentiation. It is important to elucidate the role of MGP on bone and vessel health in future.
We added the following five references in this section.
- Luo G, Ducy P, McKnee MD, Pinero JP, Loyer E, Behringer PR, Karsenty G. Spontaneous calcification of arteries and cartilage in mice lackling matrix Gla protein. Nature 1997: 386: 78-81.
- Lawton DM, Andrew JG, marsh DR, Hoyland JA, Freemont AJ. Expression of the gene encoding the matrix gla protein by mature osteoblasts in human fracture non-unions. J Clin Pathol: Mol Pathol 1999; 52: 92-96.
- Zhang J, Zhenrong Ma, Yan K, Wang Y, Yang Y, Xiang W. Matrix gla protein promotes the bone formation by up-regulating Wnt/ß-catenin signaling pathway. Front Endocrinol 2019: 10: 891-901.
- Zhang Y, Zhao L, Wang N, Li J, He F, Li X, Wu S. Unexpected role of matrix Gla protein in osteoclasts: Inhibiting osteoclast differentiation and bone resorption. Mol Cell Biol 2019: 39; e00012-19.
- Fusaro M, Gallieni M, Porta C, Nickolas TL, Khairallah P. Vitamin K effects in human health: new insights beyond bone and cardiovascular health. J Nephrol 2020: 33; 239-249.

Round 2
Reviewer 1 Report
The authors should cite those similar review papers and make a clear statement of what this review adds to the field as compared with previous similar reviews (not just reviews from Nutrients Journal, as well as other scientific journals, e.g. doi: 10.1016/s0899-9007(01)00709-2; doi.org/10.1155/2019/2069176)
Author Response
Thank you for your comment. We have listed the review papers published in Nutrients during the past around 10 years because the reviewer 1 pointed out there was no novel point of view in our manuscript compared to the previous reviews in the previous comments. We could not agree with the statement presented by reviewer 1. Because, our statement in the previous response letter does not match our context of manuscript. Therefore, we think there is no need to cite previous literature because of missing points. The reference entitled on Undercarboxylated osteocalcin: Experimental and human evidence for a role in glucose homeostasis and muscle regulation of insulin sensitivity. Nutrients 2018: 10 (7) covered the same issue with us. However, their statement was only followed by the original report (Ref No 43, 45). Thus, we believe that we have no needs to comment on this literature. This reason why we did not cite the previous reports published in Nutrients had been stated in the previous letter of response.
The reference written by Rodoriguez et al. (doi: 10.1016/s0899-9007(01)00709-2;) stated that the disparate results in fracture risk reduction under vitamin K administration were resulted in poor clinical trials. However, our expectation about the reasons obtained conflicting data were written in p11 lines 349-360 with more details. Therefore, we think that there was no need to cite the literature commended by the reviewer 1. Please read p11 lines 349-360, carefully.
 The reviewer 1 also recommended the second reference which was written by Weber P (doi.org/10.1155/2019/2069176) entitled on vitamin K and bone health in Nutrients 2001: 17: 880-887. This review was mentioned about vitamin K supplementation on fracture prevention. However, this literature was published before the publication of the meta-analyses, which had been cited by us (Ref 65-68). We could not understand why the reviewer 1 requested the older literature in which no novel comments included. Generally speaking, the literature of meta-analysis takes priority over the simple review. Therefore, we could not respond to the reviewer 1.
Reviewer 3 Report
Now, The manuscript is very good and therefore it is suitable for publication.
Author Response
Thank you for acknowledging our revised manuscript.
This manuscript is a resubmission of an earlier submission. The following is a list of the peer review reports and author responses from that submission.
Round 1
Reviewer 1 Report
There is one similar review which just was published in Nutrients (Nutrients. 2020 Mar 31;12(4)), so this review cannot provide new information to the field. Below are some specific comments:
- Page 1 line 44-45: “In this review, we will examine recent breakthroughs in the study of the biological role of osteocalcin (OC)”. Please discuss and add statements regarding “recent breakthroughs in the study of the biological role of osteocalcin (OC)”. Adding subheadings if applicable. I assumed the following paragraphs contained recent breakthroughs but described not clearly.
- Page 2 line48-49: I would recommend revising the subtitle to summarize the relationship between vitamin K and ucOC, which may provide better coherence with the following paragraph.
- Page 4 line 101: “associated with good bone health”, please provide further clarification regarding “good bone health”. Is it defined by general bone mineral density or risk of fracture or rate of bone turnover?
- Page 4 line 105-106; Is this result specific to the adolescent? As known, the adolescent is a special period before attaining peak bone mass with a high bone turnover rate and requires more nutrient intake including calcium, vitamin D, and vitamin K. Thus, please clarify if it is specific to the younger group, not other age groups.
- Page 4 Line 108-109: regarding hepatic VKDPs for blood coagulation are essential for survival. I am not clear if this is relevant to bone health or energy metabolism?
- Page 6 line 118: “Moreover, plasma vitamin K1 or MK-7 levels required to minimize the ucOC levels were highest in the ≥70-year age group, followed by the 50-69-year and 30-49-year age groups”. The older age group may have a higher risk of blood clotting or other vitamin K related vascular diseases, which may require additional vitamin K1 or MK-7 as well. Please provide explanations or adding relevant references.
- Page 7 line 142: “showed minimal bone change but…”. According to the citation (Ducy et al. [35],) “the absence of osteocalcin leads to an increase in bone formation without impairing bone resorption. But indicating that the absence of osteocalcin did not affect bone mineralization.” I recommend revising “minimal bone change”. Otherwise, it is a bit inconsistent with the statement on page 8 lines 180-181, which mentioned a close relationship between OC and osteoblastic activity.
- Page 7 line 144, “Since the OC-/- mice is considered to mimic extreme vitamin K deficiency….”. This statement did not consider other factors related to OC such as calcium and vitamin D. Please clarify how to control the effects of calcium and vitamin D, such as additional supplement calcium and vitamin D.
- Add reference regarding the statements on page 8 line 174-177.
- Page 9 line 216-223, I am not clear about what this paragraph tried to state. This is not relevant to the effect of vitamin K supplementation on bone mineral density or fractures.
Author Response
<Responses to Reviewer 1>
- Page 1 line 44-45: “In this review, we will examine recent breakthroughs in the study of the biological role of osteocalcin (OC)”. Please discuss and add statements regarding “recent breakthroughs in the study of the biological role of osteocalcin (OC)”. Adding subheadings if applicable. I assumed the following paragraphs contained recent breakthroughs but described not clearly.
Response: Thank you for your constructive suggestions. We have added the following sentences. (Revised lines 52-58)
“In this review, we will examine the impact of vitamin K on bone health through influencing energy metabolism, in the light of a recent breakthrough in the understanding of the biological role of osteocalcin (OC), which is the most abundant vitamin K dependent bone specific protein. The role of OC must be reviewed because, OC null mice [4] exhibited obesity and glucose intolerance, suggesting that OC may play an important role in energy and glucose metabolism as a bone-derived “hormone-like protein”.”
2.Page 2 line 48-49: I would recommend revising the subtitle to summarize the relationship between vitamin K and ucOC, which may provide better coherence with the following paragraph.
Response: Thank you for your helpful advice. We have revised the subtitle as follows. (Revised lines 62-63)
“Negative correlation between vitamin K status and uncarboxylated VKDPs including OC”
3.Page 4 line 101: “associated with good bone health”, please provide further clarification regarding “good bone health”. Is it defined by general bone mineral density or risk of fracture or rate of bone turnover?
Response: We agree that “good bone health” is an unclarified expression. We have thus changed “good bone health” to “bone formation”. (Revised line 127)
4.Page 4 line 105-106; Is this result specific to the adolescent? As known, the adolescent is a special period before attaining peak bone mass with a high bone turnover rate and requires more nutrient intake including calcium, vitamin D, and vitamin K. Thus, please clarify if it is specific to the younger group, not other age groups.
Response: Thank you for this important suggestion. Binkley et al. have reported on that (vitamin K intake reuired for γ-carboxylation of OC) in the adults [ref.14]. The text was thus revised as follows, citing this report. (Revised lines 132-142)
“The result of our study [24] inferred that the required vitamin K level for bone formation was 3 times higher than that for blood clotting, suggesting that the effect of vitamin K deficiency is more prominent on bone rather than on blood clotting in adolescents (Figure 3). A similar phenomenon was also reported in adults. Binkley et al. had assessed the ability of various doses of vitamin K1 to facilitate osteocalcin γ-carboxylation in healthy adults aged 19–36 y. They reported that daily vitamin K intake required for γ-carboxylation of OC was > 250 µg, and approximately 1000 µg/day would be required to maximally γ-carboxylate the circulating osteocalcin [14]. These vitamin K intakes required for γ-carboxylation of OC were higher than that for blood coagulation.”
5.Page 4 Line 108-109: regarding hepatic VKDPs for blood coagulation are essential for survival. I am not clear if this is relevant to bone health or energy metabolism?
Response: We apologize for our immature statement.
On page 4 lines 104-106, we stated that the higher vitamin K requirement for bone formation was 3 times higher than that for blood clotting. The aim of the statement was to explain that the vitamin K requirement may be different among the organs. Actually, there was no need of further elaboration in lines 108-109, because the sentence was beyond the study objectives (bone health). Therefore, we have marked this sentence (strikethrough) for deletion. Please consider this sentence as deleted. (Revised lines 143-146)
6.Page 6 line 118: “Moreover, plasma vitamin K1 or MK-7 levels required to minimize the ucOC levels were highest in the ≥70-year age group, followed by the 50-69-year and 30-49-year age groups”. The older age group may have a higher risk of blood clotting or other vitamin K related vascular diseases, which may require additional vitamin K1 or MK-7 as well. Please provide explanations or adding relevant references.
Response: Thank you for this progressive suggestion. As you have suggested, the older age group may have a higher risk of blood clotting or other vitamin K related vascular diseases. However, at a moment, it remains unclear whether the older age group require additional vitamin K1 or MK-7 for blood clotting, or other vitamin K related vascular diseases. Therefore, we have added the following sentences and citations. (Revised lines 158-164)
“In addition, plasma vitamin K1 or MK-7 levels required to minimize the ucOC levels were highest in the ≥70-year age group, followed by the 50-69-year and 30-49-year age groups (Figure 4) [11]. The older age group may also have a higher risk of vitamin K-related vascular diseases such as aortic calcification, which has been associated with the bone health [33,34]. Whether a proportion of elderly people require additional vitamin K1 or MK-7 for vascular disease remains unclear, and further investigation would be needed to confirm this.”
- Chen, Z.; Yu, Y. Aortic calcification was associated with risk of fractures: A meta-analysis. J. Back Musculoskelet. Rehabil. 2016, 29, 635-642.
- Campos-Obando, N.; Kavousi, M.; Roeters van Lennep, J.E.; Rivadeneira, F.; Hofman, A.; Uitterlinden, A.G.; Franco, O.H.; Zillikens, M.C. Bone health and coronary artery calcification: The Rotterdam Study. Atherosclerosis. 2015, 241, 278-83.
- Page 7 line 142: “showed minimal bone change but…”. According to the citation (Ducy et al. [35],) “the absence of osteocalcin leads to an increase in bone formation without impairing bone resorption. But indicating that the absence of osteocalcin did not affect bone mineralization.” I recommend revising “minimal bone change”. Otherwise, it is a bit inconsistent with the statement on page 8 lines 180-181, which mentioned a close relationship between OC and osteoblastic activity.
Response: Thank you for highlighting this. Indeed, Ducy et al. stated as you said, “ the absence of osteocalcin leads to an increase in bone formation without impairing bone resorption. But indicating that the absence of osteocalcin did not affect bone mineralization.” We have made following revision accordingly. (Revised lines 197-200)
“The histological examination of OC-/- mouse revealed increased bone formation using the double-label technique. The absence of osteocalcin led to an increase in bone formation without impairing bone resorption [4].”
- Page 7 line 144. Since the OC-/- mice is considered to mimic extreme vitamin K deficiency… This statement did not consider other factors related to OC such as calcium and vitamin D. Please clarify how to control the effects of calcium and vitamin D such as additional supplement calcium and vitamin D.
Response: Thank you for this suggestion. . (Revised lines 202-203)
We expect the animal did not have calcium, vitamin D and vitamin K deficiency, because the experiment protocol exhibit that the animal were fed by normal forage. Therefore, our original statements may make misunderstands. We deleted these sentences.[Revised lines 195-196]
- Add reference regarding the statements on page 8 lines 174-177.
Response: Thank you for your advice. We have added the citation for this statement: ref 42, Hao et al. (Revised line 262)
- Page 9 lines 216-223. I am not clear about what the paragraph tried to state…..
Response: We are emphasizing that the although there is a lack of evidence on fracture risk reduction through vitamin supplementation, the idea is biologically plausible. We have revised the sentences as follows. (Revised lines 329-335)
“It is well known that serum 25(OH) vitamin D levels are increased by the administration of native vitamin D and that serum homocysteine levels are decreased by the administration of B vitamins; similarly, serum levels of ucOC were also reduced by vitamin K administration, suggesting that vitamin B, D, and K deficiencies are treatable with corresponding vitamin administration. If a deficient vitamin is improved by the administration of corresponding vitamin, the phenotype of vitamin deficiency is also expected to recover.”
Reviewer 2 Report
This is a welcome addition as the role of vitamin K in bone is still debated. The information on osteocalcin is very timely. However some parts of the paper are not clear, especially where new ideas are introduced without enough detail.
- Purpose of paper (lines 44-47) does not match title.
- . Line 55: while tea leaves have a lot of vitamin K, the resulting tea infusion does not. See Analysis of vitamin K in green tea leafs and infusions by SPME–GC-FID. Márcia Retoa, Maria E.Figueiraab, Helder M.Filipea, Cristina M.M.Almeidaac. https://doi.org/10.1016/j.foodchem.2005.09.016
- Add country to Table 1 for all studies (only some are shown) .
- Lines 95-103 provides a derivation of a cut-off for vitamin K. The authors should elaborate more on this (as no reference is given so this is new) or delete it as in the end they give the same result as the IOM report.
- Further to point 4, the evidence that bone has a 3x higher requirement than blood clotting is not clearly shown by Figure 3. Again, the authors should provide more detail.
- Figure 4, again, shows a derived relationship that the authors have developed but without enough documentation. That older people need more vitamin K in this figure is not proof bone requires more. It could be any step that is impaired with aging.
- Section 3.1 . How is vitamin K’s involvement in energy related to bone?
- Line 159: Should this not read; vitamin K through and OC may…
- Lines 146-47 describe where data in Figure 5 come from but it is not clear what “In this report” means Is Figure 5 newly derived (and if so from what data set?) If it is taken from another paper, that would violate copyright. Either way we have not enough detail on these data.
- Lines 157-8: That OC is a bone-derived hormone is interesting but still does justify this discussion in a paper describing bone health. Table 3 is diabetes not bone health outcomes.
- Line 168: Do you mean Vitamin K Deficiency in and Bone?
- One topic that is missing is the interaction of vitamin D and vitamin K. many articles on the internet but few scientific articles address this, but the authors should make some conclusion about it. For example: van Ballegooijen AJ, Pilz S, Tomaschitz A, Grübler MR, Verheyen N.Int J Endocrinol. 2017;2017:7454376. doi: 10.1155/2017/7454376. Epub 2017 Sep 12. The Synergistic Interplay between Vitamins D and K for Bone and Cardiovascular Health: A Narrative Review.
Author Response
<Responses to Reviewer 2>
- Purpose of paper (lines 44-47) does not match title.
Response: We agree with your concern. Therefore, we have changed the statement in lines 44~47. (Revised lines: 52-58)
“In this review, we will examine the impact of vitamin K on bone health through influencing energy metabolism, in the light of a recent breakthrough in the understanding of the biological role of osteocalcin (OC), which is the most abundant vitamin K dependent bone specific protein. The role of OC must be reviewed because, OC null mice [4] exhibited obesity and glucose intolerance, suggesting that OC may play an important role in energy and glucose metabolism as a bone-derived “hormone-like protein.”
2.Line 55: while tea leaves have a lot of vitamin K, the resulting tea infusion does not. See Analysis of vitamin K in green tea leafs and infusions by SPME–GC-FID. Márcia Retoa, Maria E.Figueiraab, Helder M.Filipea, Cristina M.M.Almeidaac. https://doi.org/10.1016/j.foodchem.2005.09.016
Response: Thank you for your advice. We are aware that the resulting tea infusion does not contain a lot of vitamin K. To avoid misunderstanding, the reference to description of green tea leaf has been deleted. (Revised line 70)
3.Add country to Table 1 for all studies (only some are shown) .
Response: Thank you for your helpful advice. Countries were added for all rows of Table 1.
4.Lines 95-103 provides a derivation of a cut-off for vitamin K. The authors should elaborate more on this (as no reference is given so this is new) or delete it as in the end they give the same result as the IOM report.
Response: Thank you for this important suggestion. A reference that should have been inserted there was missing. The Reference and explanation were added to text as follows. (Revised lines 120-133)
“To address this, in our previous work, we established a new method for estimating vitamin K intake by curvature analysis using the serum levels of ucOC or PIVKA-II [24]. We had used a logarithmic regression equation obtained from vitamin K intake values and serum ucOC or PIVKA-II levels. The cutoff point was determined to be the vitamin K intake value that showed the highest curvature. As a result, in adolescents, serum ucOC and PIVKA-II levels were negatively correlated with vitamin K intake. In the curvature analysis, the vitamin K intake values associated with bone formation and normal blood coagulation were 155-188 and 62-54 µg/day, respectively. The estimated value required for blood coagulation was approximately 1 µg/day/kg body weight, which is consistent with that of previous report that provided a basis for establishing Adequate Intake (AI) values of vitamin K in the Dietary Reference Intakes (DRIs) in United States and Canada [32]. AIs of vitamin K for bone health have not been evaluated in the DRI so far, and further research is needed.”
5.Further to point 4, the evidence that bone has a 3x higher requirement than blood clotting is not clearly shown by Figure 3. Again, the authors should provide more detail.
Response: Thank you for your helpful suggestion. Figure 3 was revised as follows. (Revised lines 155-157)
Figure 3. Relationship between vitamin K nutritional status and uncarboxylated vitamin K-dependent proteins (VKDPs) as a vitamin K deficiency marker in adolescents [24].
6.Figure 4, again, shows a derived relationship that the authors have developed but without enough documentation. That older people need more vitamin K in this figure is not proof bone requires more. It could be any step that is impaired with aging.
Response: Thank you for this important suggestion. We apologize for missing the documentation part and have added the relevant citation to reference [11]. Further, we appreciate that there is an element of uncertainty involved in the second part and have accordingly revised the text as follows. (Revised lines 158-173)
“In addition, plasma vitamin K1 or MK-7 levels required to minimize the ucOC levels were highest in the ≥70-year age group, followed by the 50-69-year and 30-49-year age groups (Figure 4) [11]. The older age group may also have a higher risk of vitamin K related vascular diseases such as aortic calcification, which has been associated with bone health [33,34]. Whether a proportion of elderly people require additional vitamin K1 or MK-7 for vascular disease remains unclear, and further investigation would be needed to confirm this. Further, these results [11] suggest that older people need higher levels of circulating vitamin K levels than younger people; however, it is not clear whether this phenomenon could be attributed to increased age-related requirements, or any metabolic process that is impaired due to aging. Figure 4. Relationship between plasma vitamin K1 or menaquinone-7 (MK-7) levels and uncarboxylated osteocalcin (ucOC)-to-intact osteocalcin (OC) ratio according to age group [11].”
- Section 3.1 . How is vitamin K’s involvement in energy related to bone?
Response: Thank you for raising this concern. We have clarified this by adding information on the underlying mechanisms, as indicated by the revised subtitle of the section 3.1 “Vitamin K and cOC Deficiency may Predispose to Diabetes-related Bone Damage.” (Revised line 186) This section explained the relationship between bone and energy metabolism (Bone-Pancreas axis). This relation among the organs may be mediated by osteocalcin, which is activated by vitamin K. The following sentences were inserted. (Revised lines 190-193)
“Vitamin K deficiency reduced cOC and increased ucOC, whereas vitamin K repletion recovered these abnormalities [37]. Therefore, the model of cOC deficiency may mimic severe vitamin K deficiency, at least regarding OC metabolism.”
- Line 159: Should this not read; vitamin K through andOC may…
Response: We agree with your suggestion and changed “and” to “through”. [line 267]
- Lines 146-47 describe where data in Figure 5 come from but it is not clear what “In this report” means Is Figure 5 newly derived (and if so from what data set?) If it is taken from another paper, that would violate copyright. Either way we have not enough detail on these data.
Response: We apologize for our insufficient explanation of Figure 5. This data came from our previous report cited as Ref number 37. We are not sure whether this exhibition is a violation of copy right. On the safer side we would like to delete Figure 5 and replaced it by Table 4 that describes the cox proportional hazard model used to evaluate the association of low osteocalcin level with incidence of diabetes after adjusting for confounders. In addition, we have inserted the details on study population and revised as follows. Please consider the previous text that has been marked for deletion (strikethrough) as deleted. (Revised lines 205-248)
“We have previously investigated the relationship between serum OC and incident diabetes mellitus in a total of 1691 postmenopausal women belonging to the Nagano cohort study [39]. The mean observation period was 7.6 years and 61 cases were newly diagnosed as having type II diabetes mellitus (HbA1c was 6.5% or more, consistently) during the observation period. Table 3 indicates the age-adjusted incidence rates of diabetes mellitus with reference to the quartile with baseline OC levels. Each quartile had an insignificant difference in the observation period as expressed by person-years. As shown in the table 3, age-adjusted hazard ratios of osteocalcin quartile to incident diabetes were 1.0 (reference population), 3.25, 3.58, and 8.05 for Q4, Q3, Q2, and Q1, respectively (p for trend<0.01). Thus, the lowest-OC quartile (Q1) showed 8 times higher incidence of diabetes than Q4. Moreover, in the multiple Cox proportional hazard model, presence of low osteocalcin levels (<6.1 ng/mL) was a significant and independent risk factor for type II diabetes mellitus after adjustment of confounders such as body mass index, hemoglobin A1c level, and adiponectin-to-leptin ratio (Table 4). In mouse studies, it was demonstrated that the metabolic effects of OC on glucose homeostasis are mediated through its undercarboxylated form (ucOC) [38]. However, in women, only total OC, and not ucOC, levels were significantly associated with HbA1c [41]. These inter-species differences in the role osteocalcin molecule on glucose metabolism have not been resolved until now and further studies are required in this regard.”
Table 4 Multivariate Cox proportional hazard model for incidence of type 2 diabetes mellitus adjusting for confounders.
|
Item |
HR |
95% CI |
P |
|
Age, years |
1.049 |
1.003–1.096 |
0.037 |
|
BMI, Kg/m2 |
1.078 |
0.974–1.193 |
0.149 |
|
Osteocalcin, <6.1 ng/mL |
2.481 |
1.274–4.833 |
0.008 |
|
Triglycerides, mg/dL |
1.001 |
0.998–1.004 |
0.376 |
|
NTX, nmol BCE/nmolCr |
0.999 |
0.983–1.015 |
0.911 |
|
hs-CRP, mg/dL, |
1.227 |
0.285–5.287 |
0.784 |
|
Adiponectin/leptin ratio |
0.803 |
0.644–1.000 |
0.050 |
|
HbA1c, % |
2.518 |
1.858–3.414 |
<0.0001 |
|
L2-4BMD, g/cm2 |
1.497 |
0.306–7.330 |
0.619 |
|
Phosphate, mg/dL |
1.908 |
0.910–4.003 |
0.087 |
|
Homocysteine, nmol/mL |
1.017 |
0.918–1.127 |
0.745 |
HR; hazard ratio, CI; confidence interval, BMI; body mass index, NTX: N-teropeptide, hs-CRP; high-sensitivity C-reactive protein, L2-4BMD; Lumbar (L2-4) bone mineral density.
- Lines 157-8: That OC is a bone-derived hormone is interesting but still does justify this discussion in a paper describing bone health. Table 3 is diabetes not bone health outcomes.
Response: We agree with your opinion. However, the lower signal from bone (OC) to pancreas firstly disturbs insulin secretion or sensitivity, and can predispose to glucose intolerance or diabetes mellitus secondarily, and as a complication of that diabetes induced bone matrix protein (collagen) deterioration can occur, which can further facilitate the bone fractures. Therefore, we believe in the existence of a bone-pancreas relationship. We have added the following sentences in the manuscript to explain this negative cycle. (Revised lines 256-265)
“Therefore, vitamin K and OC may be one of the significant regulators of glucose or energy metabolism, and we believe in the existence of a bone-pancreas relationship. In summary, vitamin K deficiency in the bone results in a lower production of OC and a low serum level of OC, predisposing to a state of glucose intolerance and diabetes mellitus that may then enhance bone matrix deterioration via production of cross-linked advanced glycation end products (AGE) [42], which have been further associated with bone fractures after adjustment of confounders [43]. We believe that as a whole, vitamin K deficiency plays an important role in glucose metabolism ultimately leading to a disturbance of bone quality.” (Shiraki M, Kuroda T, Tanaka S, Saito M, Fukunaga M, Nakamura T. Nonenzymatic collagen cross-links induced by glycoxidation (pentosidine) predicts vertebral fractures. J Bone Miner Metab 26: 93-100, 2008.)
- Line 168: Do you mean Vitamin K Deficiency in and Bone?
Response: Thank you for highlighting the need to improve the clarity. We have revised the subtitle to enhance clarity as follows. (Revised line 268)
“Vitamin K Deficiency is Associated with Bone Fractures.”
- One topic that is missing is the interaction of vitamin D and vitamin K. many articles on the internet but few scientific articles address this, but the authors should make some conclusion about it. For example: van Ballegooijen AJ, Pilz S, Tomaschitz A, Grübler MR, Verheyen N.Int J Endocrinol. 2017;2017:7454376. doi: 10.1155/2017/7454376. Epub 2017 Sep 12. The Synergistic Interplay between Vitamins D and K for Bone and Cardiovascular Health: A Narrative Review.
Response: Thank you for this valuable information. We have added the following sentences. (Revised lines 351-366)
“Possible Implications of Combined Vitamin D and K Supplementation on Bone Quality:
Since the gene expression of OC is enhanced by 1,25(OH)2D following which the OC molecule receives a post-translational modification by vitamin K [69], it can be hypothesized that a synergistic or additive effect of vitamins D and K may exist in the bone. Further, an in vitro study indicated that vitamin K2 promoted 1 alpha25(OH)2D3-induced mineralization in human periosteal osteoblasts. [70] Furthermore, a study on human osteoblast cell culture demonstrated that glycoxidation interfered with maturation of osteoblasts, and this process was counterbalanced by adding vitamins D and K that reversed the detrimental effect of pentosidine, which is a well-characterized cross-linking advanced AGE. [71]. We have previously reported that multiple-vitamin deficiency carried a higher fracture risk than single-vitamin deficiency [2]. Based on these reports, it can be inferred that supplementation of vitamins D and K together might have a greater beneficial effect on bone quality than that obtained by their individual supplementation, especially during the recovery of osteoblasts from the effects of accumulated AGEs.”
Further, following additional statements were inserted into the conclusion section. (Revised lines 388-391)
“A recent in vitro study revealed that a combined vitamins D and K supplementation might help in recovery of bone quality that is otherwise not treatable by the traditional drugs for osteoporosis. This possibility should also be examined more extensively.”